



# Evaluating Turbulent and Microphysical Schemes in ICON for Deep Convection over the Alps: A Case Study of Vertical Transport and Model–Observation Comparison

Hemanth Kumar Alladi[1], Julian Quimbayo-Duarte[1], Luca Bugliaro[2], Johanna Mayer[2,3], and Juerg Schmidli[1]

[1]Institute for Atmospheric and Environmental Sciences, Goethe University Frankfurt, Frankfurt/Main, Germany
[2]Deutsches Zentrum für Luft- und Raumfahrt (DLR), Institut für Physik der Atmosphäre, 82234 Weßling, Germany
[3]ESA/ESRIN, Frascati, Italy

**Correspondence:** Hemanth Kumar Alladi (alladi@iau.uni-frankfurt.de)

**Abstract.** The Alpine region experiences frequent deep convection during summer, driven by thermal and mechanical forcing associated with the complex terrain. Deep convection transports moisture into the upper troposphere and lower stratosphere, which affects the climate through its radiative interactions. It is poorly represented in models that rely on parameterized convection and lack adequate representation of boundary layer turbulence and microphysics. In this study, we investigate the

evolution of moist deep convection observed on 8 July 2021 over the Alps using ICON simulations with explicitly resolved convection (horizontal resolution of 1 km). The simulations use two turbulence parameterizations the default turbulence kinetic energy (TKE) and the newly developed two turbulence energies (2TE) scheme combined with single moment (SM) and double moment (DM) microphysics schemes. The simulations are evaluated using cloud properties derived from MSG/SEVIRI satellite measurements. The sensitivity of cross tropopause transport to the choice of turbulence and microphysics scheme is

examined. Although, the ICON simulations capture the observed diurnal cycle of convection and successfully simulate the overshooting cloud tops during peak convective activity, our results show that the choice of the turbulence scheme influences the temporal evolution and spatial extent of deep convection, while the microphysics parameterization has a larger impact on the hydrometeor distribution and on the cross tropopause transport.

## 1   Introduction

Stratosphere–troposphere exchange (STE) processes play an important role in altering the chemical composition of the upper troposphere and lower stratosphere (UTLS) region, including radiatively active species such as ozone and water vapor (Holton et al., 1995). Through its radiative interaction, the stratospheric water vapor warms the troposphere and cools the stratosphere (de F. Forster and Shine, 1999; Shindell, 2001). The changes in the water vapor concentrations in the UTLS region can significantly affect the global climate (Solomon et al., 2010; Hegglin et al., 2014). The variations in stratospheric water vapor are

driven by processes that occur over a broad spectrum of spatial and temporal scales. Large-scale processes are well understood and accurately represented in numerical models, but the effects of small-scale processes, like deep convection, remain incom-





pletely understood and are not resolved in current global climate models (Benjamin et al., 2019; Palmer, 2014). The reliability of climate model projections depends on the accurate representation of these small-scale processes. Many observational and modeling studies reported the direct injection of water vapor into the lower stratosphere by overshooting convection (Fu et al.,

2006; Wang et al., 2009; Wright et al., 2011). The majority of these studies have focused on the tropics. The hydration into the lower stratosphere over the tropics is linked to the variability of cold-point tropopause temperatures and the confinement strength of monsoon anticyclones in the UTLS region (Mote et al., 1996; Sherwood and Dessler, 2000; Park et al., 2007; Kumar and Ratnam, 2021). The tropical upwelling associated with the lower branch of the Brewer Dobson (BD) circulation transports this water vapor into the upper stratosphere, from where it moves poleward and descends diabatically into the extratropical

lowermost stratosphere (Holton et al., 1995). The transport into the extratropical lower stratosphere from the tropical upper troposphere also occurs due to the isentropic mixing (adiabatic) associated with planetary wave activity. However, the observational studies over midlatitudes have also shown evidence of overshooting convection, which is responsible for transporting boundary layer air into the lower stratosphere (Bedka et al., 2010; Hegglin et al., 2014; Homeyer et al., 2017; Schäfler and Rautenhaus, 2023). The research on the direct injection of water vapor by overshooting convection through high resolution

simulations over midlatitudes has gained considerable interest. For example, using a 2D version of Goddard Cumulus Ensemble model, Stenchikov et al. (1996) simulated a mid-latitude mesoscale convective event and found that boundary layer tracers were directly transported into the stratosphere primarily by long convective lines. Mullendore et al. (2005) performed 10-h simulations using a 3D cloud resolving model near the Kansas-Nebraska border and reported that strong core updrafts are responsible for carrying the boundary layer tracers to the lower stratosphere. The simulations over north eastern Colorado

using a 3D non-hydrostatic cloud model with homogenous sounding and flat terrain revealed the pumping of CO, a boundary layer tracer into the upper troposphere by the supercell (Skamarock et al., 2000). Most of these studies are conducted over a flat and more or less homogeneous surfaces. However, the deep convective systems are also not uncommon over a complex terrain. The surface heterogeneities and thermal wind systems over the mountainous terrain can induce lifting due to convergence and affect cloud formation and microphysical processes (Barthlott et al., 2006; Kottmeier et al., 2008). The convective available

potential energy (CAPE) of the deep cumulus cloud activity is strongly influenced by the changes in the surface and planetary boundary layer (PBL) fluxes associated with changes in soil moisture and canopy temperatures (Segal et al., 1989; Betts et al., 1994; Findell and Eltahir, 2003; Kalthoff et al., 2011; Gentine et al., 2013; Gayatri et al., 2024).

Turbulence, an important sub-grid scale physical process in the surface and boundary layer, becomes more pronounced over complex terrain, as it can affect the development and strength of convection through the transport of moisture, heat and

momentum. The adequate representation of turbulence is important for simulating deep convective systems, which require very fine horizontal resolution. However, current global NWP and chemistry transport models cannot achieve this due to their coarse resolution. Several studies exist on the representation of PBL turbulence in NWP, such as those focusing on eddy-diffusivity closures, TKE-based schemes, and higher-order turbulence models (Mellor and Yamada, 1982; Moeng, 1984; Stull, 1988; Garratt, 1994). Some studies also reported the impact of turbulence schemes on deep convection, mainly focusing on

storm dynamics and convective system evolution (Strauss et al., 2019; Verrelle et al., 2017; Shi et al., 2019; Hanley et al., 2015). Hanley et al. (2015) found that variations in the mixing length of the subgrid-scale turbulence scheme can significantly




affect the size, intensity, and initiation time of convective storms. There are also studies which have reported the sensitivity of convection to changes in the horizontal and vertical resolutions of the model simulations (Homeyer, 2015; Jenney et al., 2023; Weisman et al., 1997; Cotton et al., 2011; Aligo et al., 2009; Keller et al., 2018; Stein et al., 2015). Using the first-

order turbulence closure, Homeyer (2015) reported that storm intensity and water vapor injection into the lower stratosphere are more sensitive to horizontal resolution than to vertical resolution. In a comparative study between cloud-resolving model simulations and radar observations conducted over southern England, Stein et al. (2015) highlighted that the simulated storm type is sensitive to the mixing length used in the turbulence parameterization. They also observed some discrepancies in storm intensity and structure and suggested further refinement in the turbulence and microphysics parameterization. The NWP and

climate models currently use bulk microphysics schemes, which assume a fixed shape for the size distribution of hydrometeors (e.g., cloud droplets, raindrops, ice crystals) to predict one or more bulk quantities (mass mixing ratio, number concentration, reflectivity factor or shape parameter) (Morrison et al., 2009). The majority of the previously mentioned cloud resolving simulations studies focused on single moment microphysics predicting only mass mixing ratio of the hydrometeors. The latest bulk microphysics schemes offer prediction of several moments for hydrometeor size distribution (Skamarock et al., 2019;

Milbrandt and Yau, 2005).

This study primarily aims to simulate overshooting deep convection over mountainous terrain using ICOsahedral Non-hydrostatic (ICON) model, focusing on turbulence parameterization and cloud microphysics. For this purpose, hindcast simulations are performed using the ICON model coupled with RTTOV (Radiative Transfer for TOVS) and evaluated using satellite observations. The main goal of this study is to compare the performance of a recently developed two-energy turbulence scheme

(2TE scheme), which is based on modeling two distinct turbulence energies, with ICON's default TKE scheme, in representing overshooting cloud tops and the vertical transport of water vapor into the UTLS. To address this and to isolate the impact of turbulence parameterizations on these processes, sensitivity experiments are performed by changing the microphysics schemes (single-moment and double-moment), increasing vertical resolution in the UTLS, and extending the model top height. These efforts aim to advance our understanding of turbulence–convection–microphysics interactions in convection-permitting models

and to contribute to the development and evaluation of the ICON framework in deep convection scenarios.

## 2  Methodology

### 2.1  Model description

The numerical simulations in the present study were carried out using the ICON model (ICON partnership (MPI-M; DWD; DKRZ; KIT; C2SM), 2024). The model solves the nonhydrostatic and compressible Navier-Stokes equations on an icosahedral-

triangular grid. Such a grid structure facilitates a seamless prediction of diverse atmospheric processes, enabling the ICON model to operate effectively across global down to local scales (Zängl et al., 2015). In this study, we used the model version icon-2024.01-dwd-2.0. The simulations use two different turbulence parametrizations. The default turbulence scheme in the ICON model developed by Raschendorfer (2001) is based on a 2nd-order closure at level 2.5 according to Mellor and Yamada (1982). The second turbulence scheme is the two-energy turbulence scheme coupled to a simplified assumed probability density



function method (2TE) (Bašták Ďurán et al., 2022). The exchange of heat, moisture, and momentum between the land surface and the atmosphere is carried out by the TERRA model, developed by Schrodin and Heise (2001) and further discussed by Heise et al. (2006). Shallow convection is parametrized using a mass-flux convection scheme (Tiedtke, 1989; Bechtold et al., 2008). The radiative scheme used is ecRad (Hogan and Bozzo, 2018), which includes an atmospheric radiative transfer model and also a radiative solver that calculates the propagation of radiation through the optical medium (Deutscher Wetterdienst
(DWD), 2019).

**Table 1.** Summary of the numerical experiments conducted in ICON.

| # | Simulation | Turb. Scheme | Microphysics | Model Top [m] | Levels |
|---|---|---|---|---|---|
| 1 | TKE-DM | Raschendorfer | Double-moment | 30 000 | 137 |
| 2 | TKE-SM | Raschendorfer | Single moment | 30 000 | 137 |
| 3 | 2TE-DM | 2TE | Double-moment | 30 000 | 137 |
| 4 | 2TE-SM | 2TE | Single moment | 30 000 | 137 |

A compilation of all simulations is listed in Table 1. The simulations ran from 2021-07-07 at 18:00:00, to 2021-07-09 at 18:00:00 UTC (Local time: CEST UTC+02:00). The initial and boundary conditions were provided by the operational COSMO-1 analysis, a product from the Swiss national weather service MeteoSwiss. The boundary conditions were provided every hour. The set-up of the simulations follows the Meteoswiss operational model, but with important modifications. The
simulation domain encompasses the greater Alpine region (Fig. 1) with a horizontal grid spacing of 1 km and a time step of 10 s. Compared to the operational configuration, the domain is a bit smaller, with the model top raised from 22500 m to 30000 m above sea level, the sponge layer from 12500 m to 20000 m and the number of vertical levels is increased from 80 to 137. A vertical grid spacing of 200 m is employed in the UTLS to better capture the processes in this region. At these resolutions, convection and the diurnal cycle are well represented without the need for deep convection parameterization (Done et al., 2004;
Pearson et al., 2014). The soil characteristics were provided by the FAO Digital Soil Map of the World (Sanchez et al., 2009), a comprehensive, geospatial database offering detailed information on soil types, properties, and characteristics worldwide. The orography used the Advanced Spaceborne Thermal Emission and Reflection Radiometer dataset (ASTER, Yamaguchi et al., 1998). The dataset typically offers a resolution of around 30 meters. The datasets have been interpolated to match the resolution of the numerical domain which has a horizontal grid spacing of 1 km (Fig. 1).
The ICON model employs the fast radiative transfer model (RTTOV) as the forward operator to simulate top of atmosphere radiances and brightness temperature from the atmospheric state (Saunders et al., 2018). Cloud processes were parametrized using a single-moment (Seifert, 2008; Doms et al., 2011) and double-moment (Seifert and Beheng, 2006) micro-physics (Table 1). The ICON single-moment scheme predicts five hydrometeor classes: cloud water, rainwater, cloud ice, snow, and graupel. Graupel forms through riming, where ice or snow particles collide with supercooled liquid droplets. Most microphysical pro-
cesses are highly sensitive to particle size. While the mean particle size is often correlated with mass content, this correlation is not always reliable. Double-moment schemes address this limitation by predicting both the mass content and the number



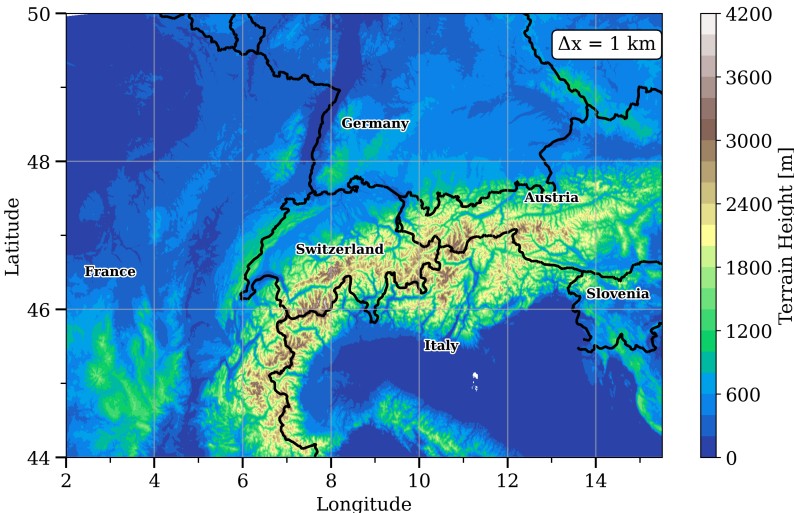

**Figure 1.** Topographic map of the ICON domain (colour shading) with a horizontal grid spacing of 1 km.

concentration of hydrometeors. By providing independent size information, these schemes offer a more accurate representation of particle size distributions. The double-moment option refers to the prediction of both mass content and number concentration, which are statistical moments of the particle size distribution. The double-moment microphysics scheme predicts the

specific mass and number concentrations of cloud water, rain water, cloud ice, snow, graupel and hail. This scheme is suitable at convection-permitting or convection-resolving scales, i.e., mesh sizes of 3 km and finer. Only on such fine meshes the dynamics is able to resolve the convective updrafts in which graupel and hail form. As for convection, only the shallow convection parametrization is set on for the TKE simulations.

## 2.2  Observations

Meteosat Second Generation (MSG, Schmetz et al. (2002)) comprises four geostationary satellites (Meteosat-8 to Meteosat-11) launched between August 2002 and July 2015. The main payload in the MSG satellite series is the Spinning enhanced visible and infrared Imager (SEVIRI). SEVIRI is a line-by-line optical scanning radiometer that provides continuous global images every 15 minutes in 12 spectral bands (4 visible and near-infrared (VNIR) and 8 infrared (IR) channels) covering a spatial range from 80°E to 80°W and 80°S to 80°N, which includes Africa, Europe, and a large portion of the Atlantic Ocean. The

spatial resolution at nadir for the 11 channels, ranging from 0.6 μm to 14 μm at nadir, is $3 \times 3$ km$^2$, whereas it is $1 \times 1$ km$^2$ for the high resolution visible channel. The IR channels centered at 10.8 μm and 12 μm are used to estimate surface and cloud top temperatures. The water vapor and winds in the atmosphere are retrieved from the radiances received in the broad absorption channels at 6.2 μm and 7.3 μm. The visible channel is used to retrieve cloud microphysics.For details regarding the principle and the retrieval of different satellite products, the readers are referred to (Schmetz et al., 2002).



## 2.3 Detection of overshooting convection and ice water path

The most common method to identify the overshooting cloud top is the Brightness Temperature Difference (BTD) method (Fritz and Laszlo, 1993; Schmetz et al., 1997; Bedka et al., 2010). In this method, the difference of the brightness temperature (BT) values between the water vapor channel (WV) at 6.2 μm and the IR atmospheric window channel at 10.8 μm is calculated and is represented as $BT_{(6.2\,\mu m - 10.8\,\mu m)}$. The $BT_{(6.2\,\mu m - 10.8\,\mu m)}$ is positive above deep convective overshooting clouds due to the presence of water vapour above the cloud top (Schmetz et al., 1997; Setvák et al., 2007). The water vapour present above the cloud top can emit more radiation if the temperature above the cloud top increases, leading to a warmer brightness temperature in the WV channel. The cloud top, being at a relatively lower temperature, emits less radiation, resulting in a colder brightness temperature in the IR channel, where water vapour absorption is very low. Warmer temperatures or higher water vapour amounts in the lower stratosphere above the cloud top can result in a larger positive $BT_{(6.2\,\mu m - 10.8\,\mu m)}$. The maximum difference in BTs between the WV and IR channels can range from 4 to 8 K (Schmetz et al., 1997; Setvák et al., 2007). Setvák et al. (2007) suggested that every overshooting cloud top can generate moisture, which rapidly balances its temperature with its warmer surroundings. In this study, SEVIRI data have been used for detecting overshooting cloud tops.

The Cirrus Properties from SEVIRI algorithm is used to retrieve ice cloud cover (all clouds with icy tops) as well as ice optical thickness ($\tau$) and ice water path (IWP) for thin cirrus (Strandgren et al., 2017a). CiPS is an AI-based algorithm trained with thermal SEVIRI observations and coincident cirrus properties retrieved from the Cloud-Aerosol Lidar with Orthogonal Polarization (CALIOP) instrument (Winker et al., 2009). CiPS is highly sensitive to thin cirrus but the thermal observations saturate for thick clouds in deep convection. Thus, CiPS is very well suited to derive the spatial extension of the convective clouds including their anvils, but for thicker clouds, the Algorithm for the Physical Investigation of Clouds with SEVIRI (APICS) is better suited (Bugliaro et al., 2011). APICS exploits two SEVIRI solar channels centered at 0.6 and 1.6 $\mu$m to derive the optical thickness and effective radius ($r_{\mathrm{eff}}$) with a Nakajima-King-like method (Nakajima and King (1990)) under the assumption of ice aggregates from Baum et al. (2014).

IWP is derived under the assumption of a vertically homogeneous ice cloud layer with $IWP = 2/3 * \rho_{\mathrm{ice}} * r_{\mathrm{eff}} * \tau$, where $\rho_{\mathrm{ice}} = 917\,\mathrm{kg\,m^{-3}}$ is the density of ice. Since optical thickness ranges up to 200 and effective radius between 5 and 60 $\mu$m, maximum retrieved IWP is around 7300 g m$^{-2}$. IWP from this VIS-NIR retrieval is expected to be rather sensitive to cloud ice particles and has been shown to be similar or underestimate IWP from microwave and active observations (Rybka et al., 2021),which are sensitive to larger hydrometeors. Due to its dependence on solar observations, APICS is limited to daytime, whereas CiPS is available both during daytime and at night (for details on the algorithms, see Strandgren et al. (2017a, b); Bugliaro et al. (2011); Rybka et al. (2021)). Thus, in this study, ice-cloud cover is retrieved from CiPS, while IWP is taken from APICS (daytime only). The algorithms are applied to rapid-scan data from MSG-3 (Meteosat-10) with a temporal resolution of 5 min. On 8 July 2021 the day under consideration, two data gaps occur at 11:40–12:00 UTC and 21:10–21:15 UTC



## 3 Results

### 3.1 Synoptic overview

In the present study, potential vorticity (PV), horizontal winds (zonal wind-u and meridional wind-v), pressure vertical velocity ($\omega$) and geopotential height from the ERA-5 reanalysis, available at a 1-hour temporal resolution and a $0.25° \times 0.25°$ horizontal resolution, are used to investigate the background synoptic-scale conditions in both the lower and upper troposphere.

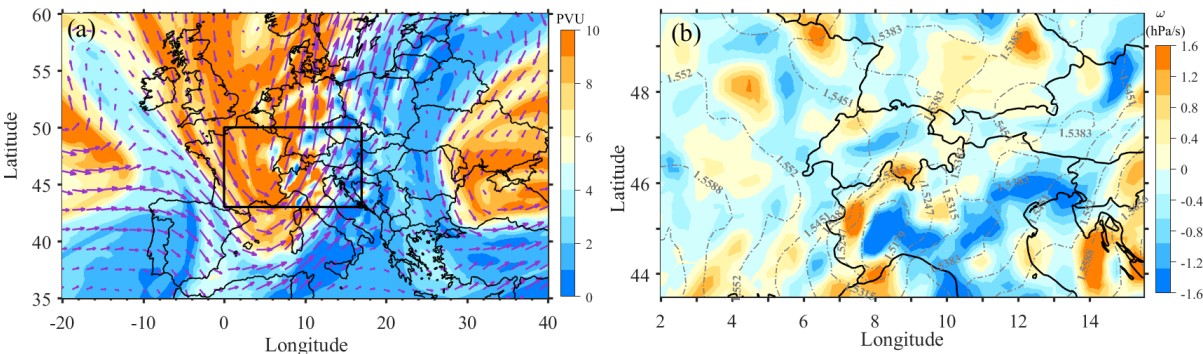

**Figure 2.** (a) Spatial distribution of potential vorticity (PV) along 350 K isentropic surface obtained on 2021-07-08, 15:00 UTC from ERA-5 reanalysis. The purple arrows represent the wind vectors. (b) Spatial distribution of pressure vertical velocity at 850 hPa obtained for the same time over the rectangular box region shown in (a), the dashed contour lines represent the geopotential height contours.

The 350 K isentropic surface shows a PV intrusion extending from higher latitudes to lower latitudes, covering Western and Central Europe, including France, Germany, and parts of Eastern Europe (Fig. 2). A clear cyclonic circulation is also observed from the wind circulation at this level, suggesting the presence of an upper-level trough over this region. The levels within and below the positive (cyclonic) upper-level PV exhibit a less stable potential temperature distribution (Hoskins et al., 1985; Thorpe, 1985). The PV anomalies and cyclonic circulation in the upper levels can destabilize the lower troposphere, potentially triggering deep convection over the regions ahead of the PV intrusion (Kiladis, 1998; Waugh and Funatsu, 2003). The 850 hPa geopotential height contours show a closed circulation over Central Europe, particularly near the European Alps, indicating the presence of a mesoscale low. This feature is evident by the presence of strong vertical updrafts (negative $\omega$ values), indicating strong convective active activity. The PV intrusion, associated with Rossby wave breaking, interacts with the upper-level cyclonic flow, leading to reduced static stability, enhanced upper-level divergence, and the facilitation of large-scale ascent and deep convection (Kiladis, 1998). The upper-level PV cyclonic low together with the mesoscale low in the lower troposphere can further enhance the strength of the convection, thereby leading to deep overshooting convection.





### 3.2    Evaluation of ICON simulations with satellite observations

#### 3.2.1    Cloud-top temperature and diurnal cycle of convection

Fig. 3 compares the simulated BT at 10.8 μm with the SEVIRI observations on 08 July 2021 at 15:00 UTC, representing the late afternoon hours when convection over the land is near its peak intensity (Chen and Houze Jr, 1997; Yang and Slingo, 2001). Lower BTs indicate colder cloud tops, while warmer BTs correspond to low-level clouds or clear skies or thin cirrus. Satellite observations show very low BTs ($< 240$ K) mainly located over northern Italy, and southern Germany, eastern Switzerland, and western Austria. The simulations also capture this behavior, but with slight differences in their spatial extent and location.

To understand the diurnal evolution of convection, we analyze the temporal patterns of BT in the domain (Fig. 1) and select a region (dashed box in Fig. 3) that exhibited deep convection which developed and persisted locally, with no advection of clouds into this region from surrounding areas throughout the convective event.

During fair weather conditions, the diurnal cycle of convection is primarily driven by the surface heating and boundary layer dynamics (Weckwerth et al., 2011; Yang and Slingo, 2001; Noel et al., 2018; May et al., 2012). However, there are various
mesoscale processes like anabatic winds near orography and land sea breeze circulation near coastal regions that can also affect the diurnal cycle of convection (Basu, 2007). The presence of a cutoff low in the upper troposphere, formed as narrow filaments from stratospheric potential vorticity (PV), as seen in this study, can also have a major impact on the triggering and evolution of the deep convection (Röthlisberger et al., 2022). The diurnal evolution of convection in simulations and observations is shown in Fig. 4, which is obtained by averaging the BT (10.8 $\mu$m) in the analysis region (44.75°–47.75°N, 5.85°–10.5°E). It is clear
from this figure that the simulations and observations show a clear diurnal cycle of convection. During the night and early morning (00:00-09:00 UTC), higher BTs between 260 and 270 K with low fluctuations are found in simulations, indicating negligible or low convective activity. In the same time period, the observations show similar BT values; however, a small dip to 260 K between 04:00 and 07:00 UTC is likely caused by the advection of high clouds into this region during this period (not shown). A significant drop in BT is observed during the late morning and early afternoon (09:00-12:00 UTC) showing
the onset of convection. The lowest BT values indicating maximum convective intensity are found during the late afternoon to evening hours (14:00-16:00 UTC), after which the BT values increase due to the decay of convective activity.

A sharp decrease in the BT is visible during 12:00-16:00 UTC, indicating maximum convective activity during this period. Both TKE simulations (TKE-SM & TKE-DM) match the observed timing of peak convection, reaching their minimum BT at 15:00 UTC. While the 2TE-SM and 2TE-DM simulations exhibit a time lag of one hour (16:00 UTC) for the peak convective
activity. Signs of decay are clearly seen after 15:00-16:00 UTC; however, the simulations exhibit a slower decay of convection compared to the observations, as indicated by a slower rise in BT.

#### 3.2.2    Overshooting convection and ice clouds

To investigate the occurrence of overshooting convection, we follow the BTD method. Fig. 5 shows the difference in BTs obtained from satellite observations and ICON simulations at 15:00 UTC. Colder BT (Fig. 3), along with high positive
$BT_{(6.2\,\mu m - 10.8\,\mu m)}$, are particularly observed in the southern, northwestern, and, to some extent, northeastern regions of the se-



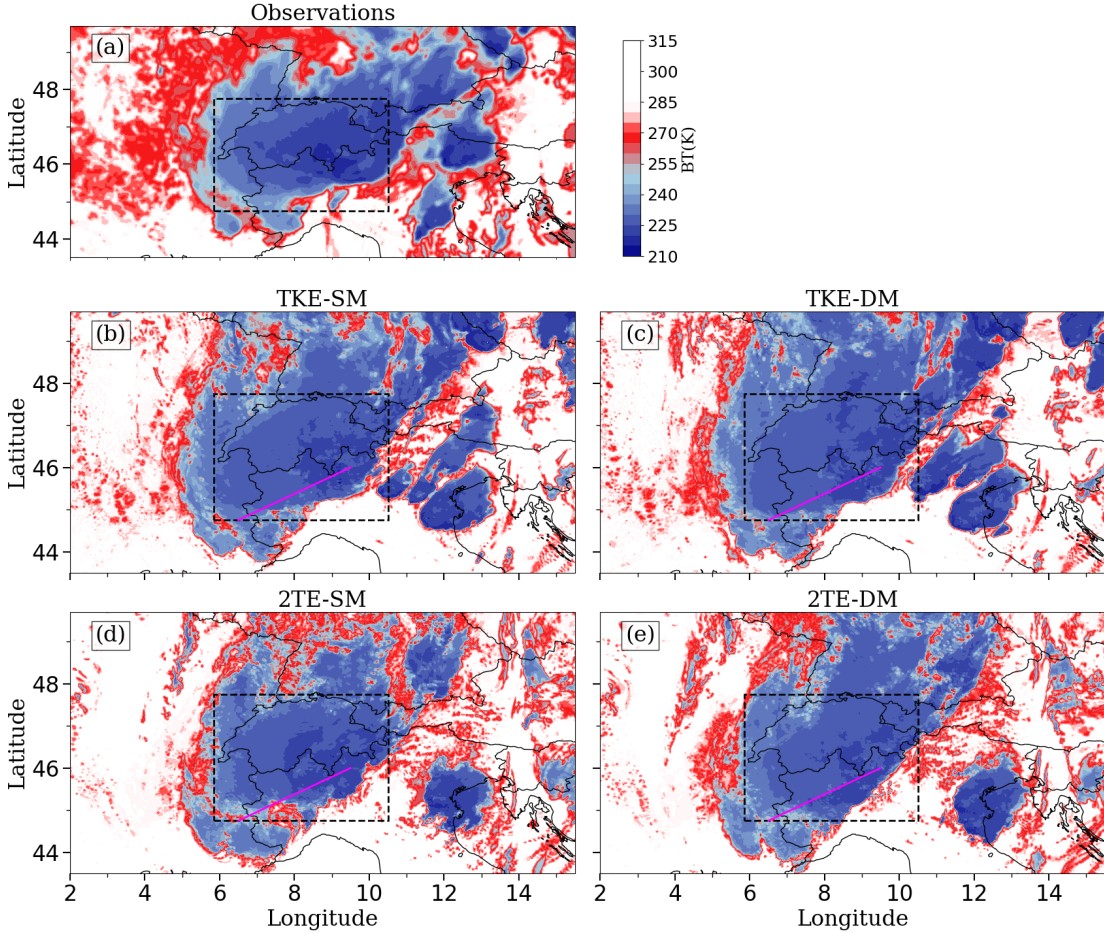

**Figure 3.** (a), Spatial distribution of BT obtained at 10.8 $\mu$m on 2021-07-08, 15:00 UTC (colour shading). (b), (c), (d) and (e) also represent the same but for ICON simulations. The box in all the panels represents the analysis region. Vertical cross-sections are obtained along the magenta line shown in (b), (d), (e) and (f).

lected region in the satellite observations. The simulations broadly replicate the positive values seen in the observations for these regions, with slight differences in terms of spatial extent, intensity, and location of the maximum positive $BT_{(6.2\,\mu\mathrm{m}-10.8\,\mu\mathrm{m})}$. High $BT_{(6.2\,\mu\mathrm{m}-10.8\,\mu\mathrm{m})}$ (>3 K) values are concentrated over the central Alpine region in both observations and simulations. The spatial coverage of the moderate $BT_{(6.2\,\mu\mathrm{m}-10.8\,\mu\mathrm{m})}$ values (1-3 K) is larger in simulations compared to observations. The

simulations exhibit a broader distribution beyond the selected region, extending eastward and southeastward in continuous or semi-continuous streaks, in contrast to the observations, which show isolated patches.

The fractional area coverage of overshooting cloud tops in the selected region, represented as $BT_{(6.2\mu m-10.8\mu m)} > 0$, ob-tained from observations and simulations, is shown in Fig. 6. Similar, to the diurnal cycle of deep convection, a strong diurnal



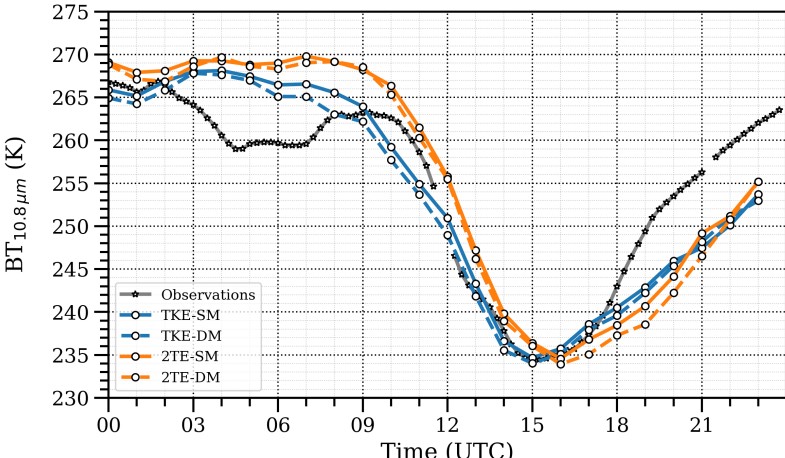

**Figure 4.** Time series of BT values at 10.8 $\mu$m for ICON simulations and satellite observations observed on the same day over the box region (44.75°–47.75°N, 5.85°–10.5°E), as shown in Figure 3.

variation in overshooting cloud tops exists in observations and simulations, with a peak between 12:00 and 16:00 UTC (after-
noon to evening local time), after which it begins to decline. The percentage of the overshooting clouds is minimal early in the day (before 06:00 UTC). It can be noted from this figure that the observations showed two distinct peaks, at 08:00-10:00 UTC and 14:00-15:00 UTC. Similar peaks with differences in their strength (magnitude) and timings are also observed in the model simulations. Despite the presence of a stable surface layer during the early morning hours, the existing PV intrusion can provide upper-level instability and dynamic lifting, thereby promoting strong convection. The overshooting cloud fraction is
further amplified (Fig. 6) during the local evening hours ($\sim$14:00-16:00 UTC) due to the peak associated with diurnal cycle of surface heating.

    The TKE simulations show larger overshooting cloud fraction compared to both observations and 2TE simulations during this period. The SM scheme uses a fixed size distribution of hydrometeors, which limits the variability in the growth process of the hydrometeors. In general, the auto-conversion and accretion processes are prevalent near the cloud top of deep convective
clouds (Houze Jr, 2014; Tao and Moncrieff, 2009). The inflexible hydrometeor assumption in the SM scheme could potentially result in enhanced BT difference due to the rapid injection of ice above the tropopause by strong convective updrafts associated with deep convection (Doms et al., 2011; Seifert, 2008). However, the DM scheme which allows the dynamic adaptation of the hydrometeor distribution (ice particle) through auto conversion (Seifert and Beheng, 2006) also shows a peak in the $BT_{(6.2\,\mu\mathrm{m}-10.8\,\mu\mathrm{m})}$ values with a magnitude larger than observed in SM. The peak in the overshooting cloud fraction is broader
in both simulations compared to observations. This is clearly evident in the TKE simulations but not in the 2TE simulations, indicating differences in the model's representation of local convective processes. This could be partly influenced by the finer horizontal resolution (1 km × 1 km), which allows the identification of localized overshooting events that may not be resolved in the relatively coarser satellite observations (3 km × 3 km).





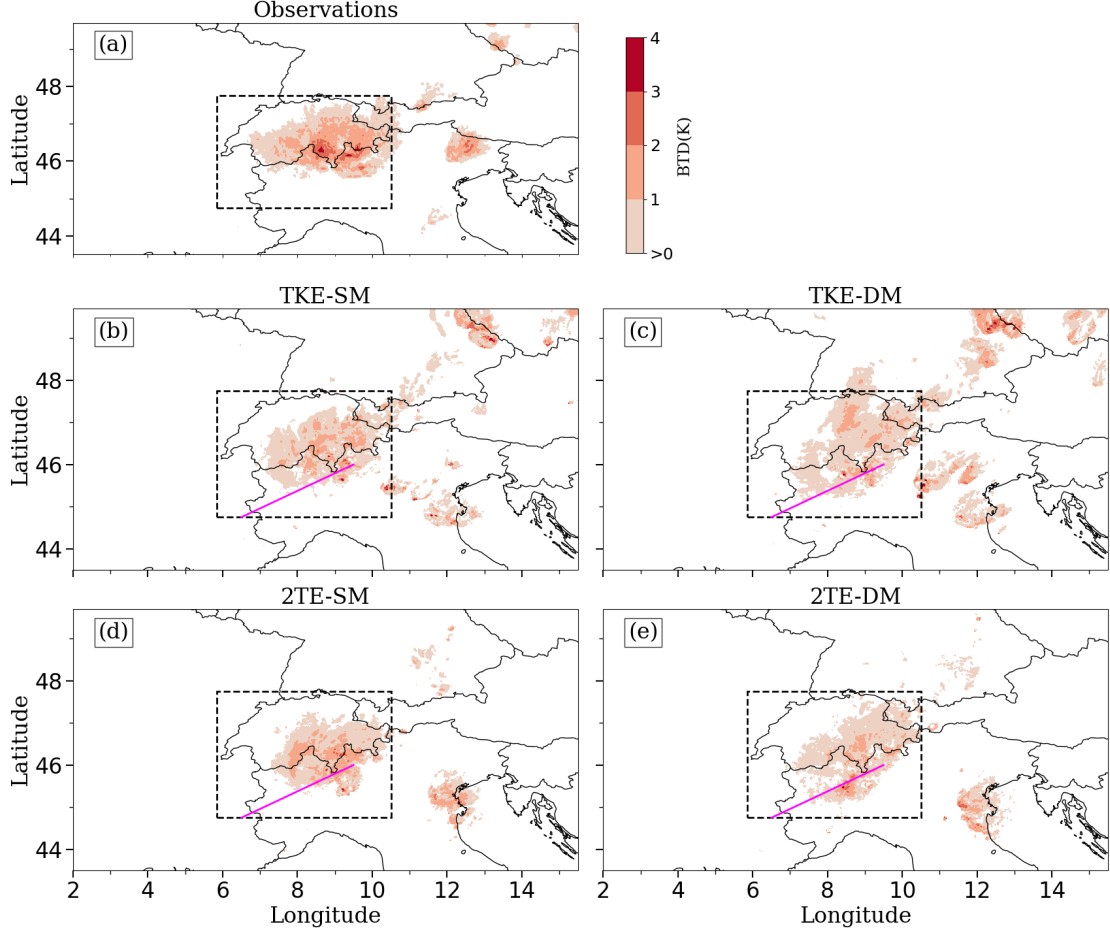

**Figure 5.** (a) Spatial distribution of the $BT_{(6.2\,\mu m - 10.8\,\mu m)}$ obtained between WV (6.2 $\mu$m) and IR (10.8 $\mu$m) at 15:00 UTC from observations, (b), (c), (d) and (e) represent the same but for ICON simulations. Vertical cross-sections of QI (cloud ice), QC (cloud water), w (vertical wind) and turbulent kinetic energy (tke) are obtained along the magenta line shown in (b), (d), (e) and (f).

Homeyer (2015) reported that the 1 km cloud-resolving simulations, performed for a convective squall line over the U.S.
Central Great Plains extending from Texas to Oklahoma, captured more overshooting cloud tops than the coarser 3 km simulations and radar observations, which have a resolution of ∼2 km. All ICON simulations show a higher fraction of overshooting cloud tops compared to observations after the peak convective activity (15:00–16:00 UTC). The DM scheme simulations show a slower dissipation compared to the SM configuration, indicating prolonged retention of convective activity.

Cloud ice constitute a large portion of the hydrometeors in overshooting deep convective systems in the upper troposphere.
One important ice cloud parameter, which is also an essential climate variable, is the ice water path (IWP) (Zemp et al., 2021). The IWP represents the vertically integrated cloud ice water content, encompassing all types of ice particles. In this study,





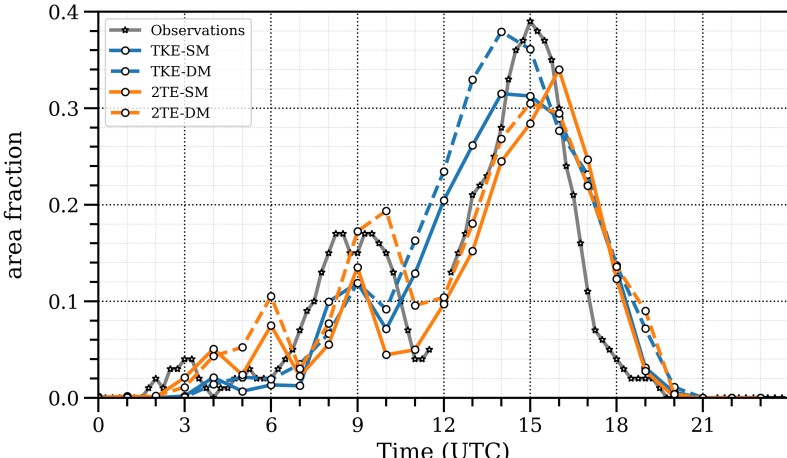

**Figure 6.** Time series of area fraction of $BT_{(6.2\mu m - 10.8\mu m)} > 0$ for ICON simulations and satellite observations obtained on 2021-07-08 over the study region.

IWP is selected for comparison as it can be retrieved from satellite observations using visible channels, in contrast to quantities such as ice water content, which are available in the model output but cannot be observed by passive satellite sensors. The diurnal evolution of the IWP over the analysis region between simulations and observations is represented in Fig. 7. The IWP

peaks around 15:00-16:00 UTC in the simulations similar to that found in observations. The TKE-DM shows a slight higher IWP compared to observations and the 2TE simulations. The IWP obtained from 2TE-SM is lowest. The use of DM shows an increase in the IWP values for both the TKE and the 2TE simulations. A recent study conducted by Gettelman et al. (2024) reported that the higher IWP in the DM scheme is attributed to the enhanced ice nucleation, reduced sedimentation rates of smaller ice particles, and accretion of ice onto snow throughout the depth of the column.

**3.3 Impact of model setup on cross-tropopause transport and cloud ice distribution**

To investigate the sensitivity of model set up on moisture transport across the tropopause, the cloud ice content (QI) above the cold point tropopause (CPT), is considered, where CPT is defined as the altitude of minimum temperature in the troposphere (Highwood and Hoskins, 1998), and $QI_{CPT}$ is the integrated ice content in a 1 km deep layer above the CPT. The diurnal evolution of $QI_{CPT}$ averaged over the analysis region is shown in Fig. 8.

In all configurations, $QI_{CPT}$ reaches a maximum between 14:00 and 15:00 UTC. This coincides with the peak in overshooting cloud fraction in both observations and simulations (Fig. 6). The DM simulations show higher $QI_{CPT}$ values and stronger vertical transport across the tropopause compared to single-moment (SM) simulations, likely due to differences in the representation of microphysical processes between them. The SM scheme predicts only the mass mixing ratio of the hydrometeor with a fixed particle size distribution. In contrast, the DM scheme prognostically simulates both the mass and number concen-

trations of hydrometeors, enabling a more realistic and dynamic representation of particle size distributions and microphysical





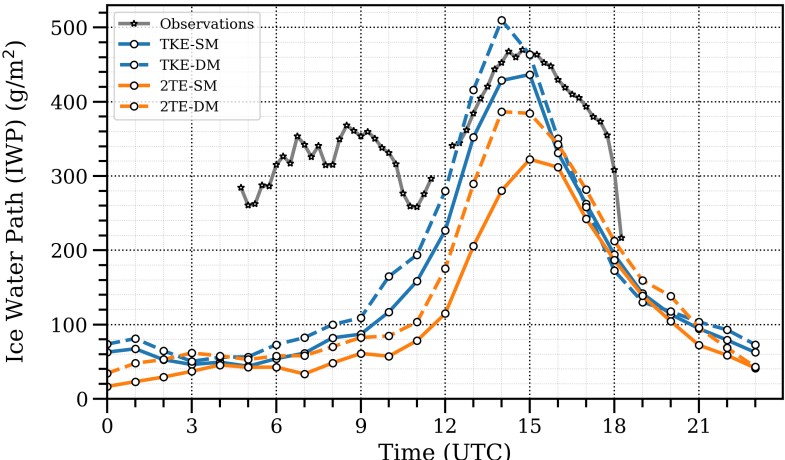

**Figure 7.** Time series of the Ice Water Path (IWP) between ICON simulations and satellite observations obtained on 2021.07.08 over the study region.

processes such as nucleation, growth, sedimentation, and sublimation. These improvements can significantly influence the vertical transport of the hydrometeor, as the DM scheme provides a more detailed representation of phase changes and associated latent heat release, which enhance buoyancy and support deeper convective updrafts. Additionally, the DM schemes tend to produce a larger number of smaller ice particles near and above the tropopause. These particles sediment more slowly and

can remain suspended for longer periods, leading to increased ice content in the upper troposphere and lower stratosphere, particularly above the CPT. This highlights the role of choice of microphysics configuration in determining the magnitude of transport into the lower stratosphere.

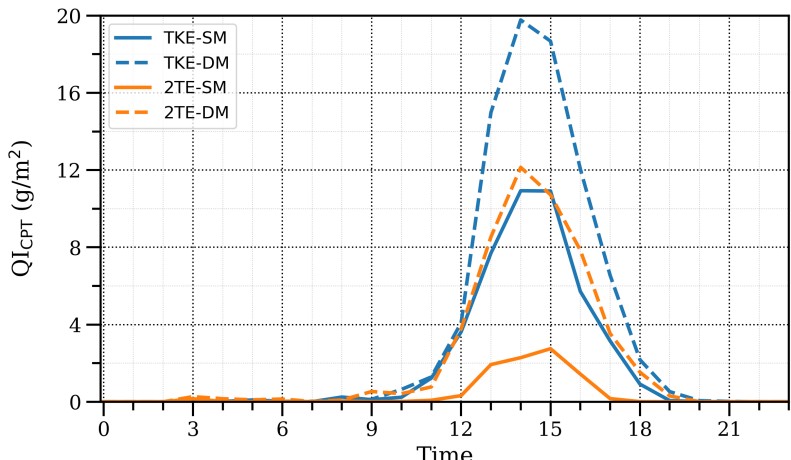

**Figure 8.** Time series of $QI_{CPT}$ averaged over the selected region on 2021.07.08 across different simulations.



To understand the regional heterogeneity and spatial extent in deep convective transport, the spatial distribution of $QI_{CPT}$ in different simulations for 15:00 UTC is shown in Fig. 9. The values are higher over regions with Alpine topography, indicating
the influence of orographic lifting on cross-tropopause transport. From this figure, it can be observed that $QI_{CPT}$ is larger in the TKE simulations, showing a higher magnitude and a broader distribution compared to the 2TE simulations. Furthermore, the addition of the DM microphysics configuration increases the vertical transport across the tropopause, leading to higher $QI_{CPT}$ values in both the TKE and 2TE simulations. The DM simulations show more intense and frequent regions of cloud ice above the tropopause, with values exceeding $200 \text{ gm}^{-2}$. While such features are also evident in the TKE-SM simulation,
they are less frequent in the 2TE-SM configuration. This suggests that the TKE simulations support cross-tropopause transport more effectively compared to the 2TE simulations which could be due to differences in the intensity of convection and mixing represented in the TKE and 2TE schemes.

To investigate this, convective available potential energy (CAPE) and vertical wind at 500 hPa averaged over the analysis region are examined. The time series of the mean, 95th percentile and maximum of these parameters for both SM and DM
configurations in the TKE and 2TE simulations are shown in Figure 10. In contrast to the 2TE simulations, the TKE simulations exhibit higher CAPE (Fig. 10 (a)-(c)) and stronger vertical wind at 500 hPa (Fig. 10 (d)-(f)), indicating greater convective instability and enhanced updraft strength, which contribute to the increased transport into the UTLS region. The DM simulations show a moderate increase in CAPE and updrafts strenght compared to SM. So, convective and complex microphysical processes near the cloud top can influence the amount of $QI_{CPT}$.

To investigate the role of turbulence, vertical cross sections of turbulent kinetic energy were obtained along a line passing through the main convective cores in the analysis region and is shown in Fig. 11. It is evident from this figure that the 2TE scheme exhibits lower turbulent kinetic energy compared to the TKE simulations, indicating weaker mixing. The close spacing of potential temperature levels near tropopause is clearly observed in the TKE simulations, indicating enhanced vertical mixing due to the updrafts (Fig. 12 (a)-(b)) associated with overshooting convection. Higher TKE values near and above the tropopause
in these simulations could be associated with gravity wave activity. Thus, the weaker mixing and reduced convective instability in 2TE simulations likely limited the vertical transport compared to TKE simulations. However, within 2TE simulations, the DM scheme shows higher transport to the lower stratosphere compared to the SM simulations, possibly because of the better representation of the microphysical processes in DM compared to SM.

Horizontal advection in the deep convection anvil region can cause a temporal lag between the maximum convective activity
and the amount of ice observed near the tropopause (Auguste and Chaboureau, 2022). Similar to turbulent kinetic energy, the vertical distribution of cloud ice and cloud water content in different simulations is also obtained and shown in Fig. 12. Higher cloud ice values are present in all the simulations in the altitude range of ∼8-12 km. It can be observed from this figure that the 2TE-SM simulation does not show any overshooting at this particular location and time, while the implementation of DM clearly enhances the vertical transport of ice particles above CPT. The DM scheme simulations show sustained regions of cloud
ice above the CPT over a larger horizontal area. This could be partly attributed to the suspension of smaller ice particles due to slower sedimentation rates and stronger updrafts. Higher cloud water values extending from the mid-troposphere to the upper troposphere are found over regions with stronger updrafts (Fig.12), indicating the potential for moisture transport into the lower





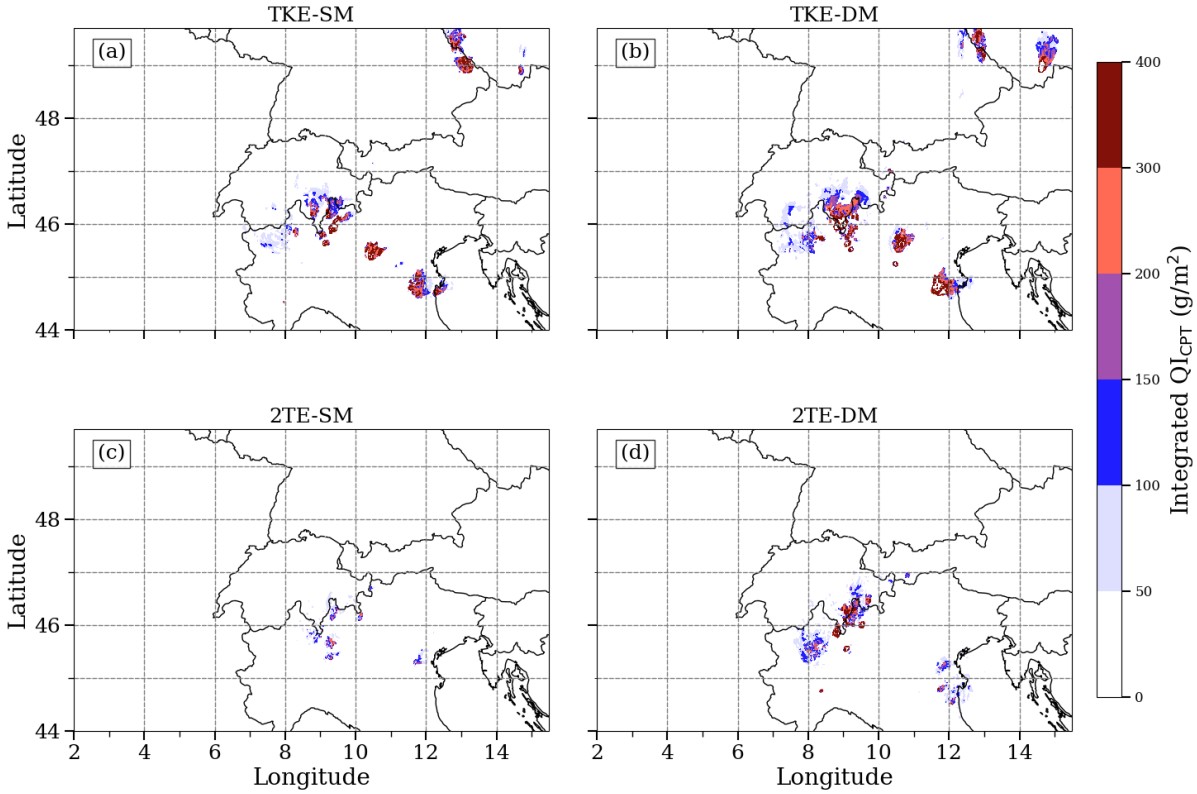

**Figure 9.** Spatial distribution of $QI_{CPT}$ obtained at 15:00 UTC on 2021.07.08 from ICON simulations.

stratosphere. It can be found from this figure that the DM simulations produce higher cloud water values compared to the SM simulations possibly due to the better representation of hydrometeor mixing processes in DM compared to SM..

The CPT appears more distorted in the DM simulations compared to SM, likely due to enhanced convective activity and latent heat release, which are likely less accurately represented in the SM scheme. To summarize, the DM schemes produce more cloud water and cloud ice compared to SM in both the TKE and 2TE simulations, indicating that the concentration of the hydrometeors and their distribution depend primarily on the choice of the turbulence parameterization.

## 4   Conclusions

This study evaluates the impact of turbulence and microphysics parameterization on the diurnal cycle of convection and the characteristics of overshooting convection in ICON simulations for a deep convective event observed on 8 July 2021 over complex terrain. Satellite observations are utilized to assess the representation of the simulated deep convection, ice clouds, ice water path and overshooting cloud tops.





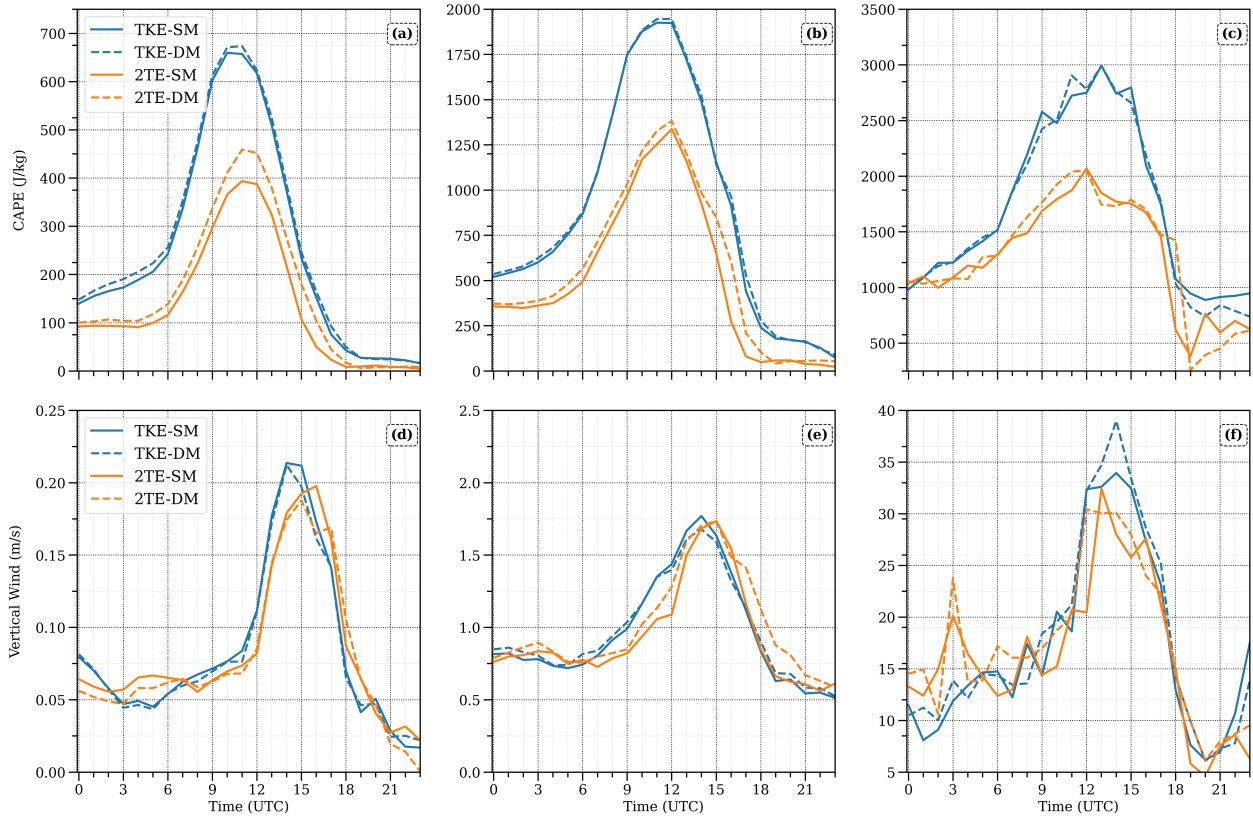

**Figure 10.** Time series of CAPE and 500 hPa vertical wind from ICON simulations over the box region (44.75°–47.75°N, 5.85°–10.5°E), as shown in Figure 3 (a)–(c) show the mean, 95th percentile and maximum of CAPE; panels (d)–(f) show the same statistics for vertical wind.

The diurnal cycle of convection is evaluated by comparing the Brightness temperature (BT) values at 10.8 $\mu$m from model
simulations with satellite observations. The assessment of the model simulations focused on the diurnal evolution of the overshooting cloud fraction and ice clouds, while the spatial evaluation examined the distribution of the convection and overshooting cloud tops within the domain region. For identifying the overshooting cloud tops, the brightness temperature difference (BTD) method is used.

The TKE and 2TE simulations effectively captured the diurnal cycle of convection, with the magnitude of peak convective
activity (indicated by low BT values) closely matching the observations. However, differences remain, particularly in the timing of peak convective activity and the evolution of the overshooting cloud fraction, highlighting the influence of the turbulence parameterization used. The TKE simulations show a larger overshooting cloud fraction, higher IWP, and greater moisture transport into the lower stratosphere than the 2TE simulations. This enhancement is likely due to stronger convection, as indicated by higher CAPE in TKE simulations compared to 2TE, larger maximum vertical velocities and stronger turbulence
due to higher turbulent kinetic energy. Thus, the main impact of the choice of the turbulence scheme on cloud and moisture




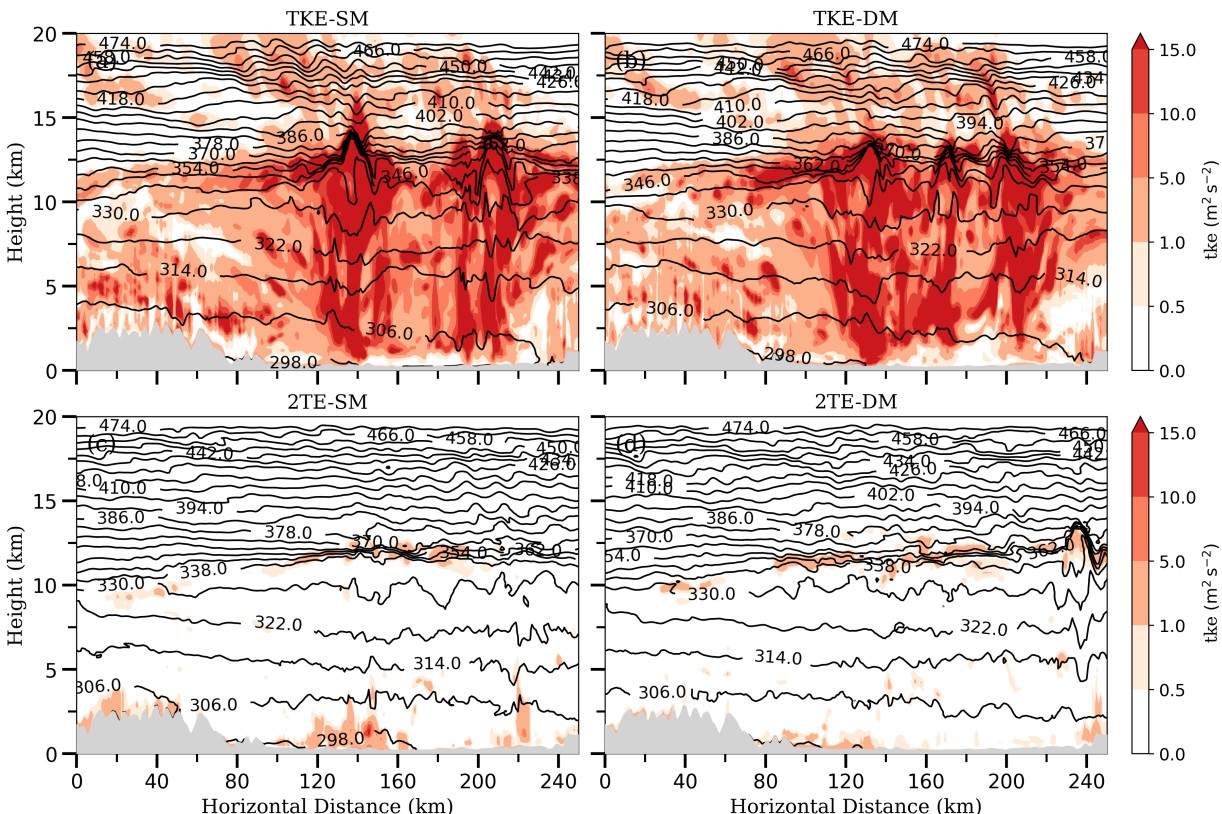

**Figure 11.** Vertical Cross section of turbulent kinetic energy (m$^2$s$^{-2}$), along the magenta line as shown in Figure 3 and Figure 5 at 14:00 UTC.The black contour lines represent the potential temperature.

distribution in the UTLS region is likely an indirect one; Its impact on the characterization and evolution of the ABL being more important than its impact on mixing in the deep convection.

There are notable differences between Double Moment (DM) and Single Moment (SM) schemes, particularly in the intensity of convection and the fraction of overshooting cloud tops. The DM scheme in both TKE and 2TE simulations, produces larger
BT differences and ice water path compared to SM, indicating that the strength and vertical extent of deep convection depends on the choice of microphysics. A recent study conducted by Gettelman et al. (2024) over the Southern Ocean highlighted that the DM scheme significantly enhances the representation of supercooled liquid and ice clouds by predicting both the mass and number concentrations of hydrometeors, leading to higher ice water path (IWP) values compared to the SM scheme.

Further, the sensitivity of the model setup on moisture transport into the lower stratosphere above the tropopause is also
examined by estimating the integrated ice content above 1 km from the cold point tropopause. It is larger using the DM scheme than the SM scheme for both the TKE and 2TE simulations. This indicates significant differences in the vertical distribution of





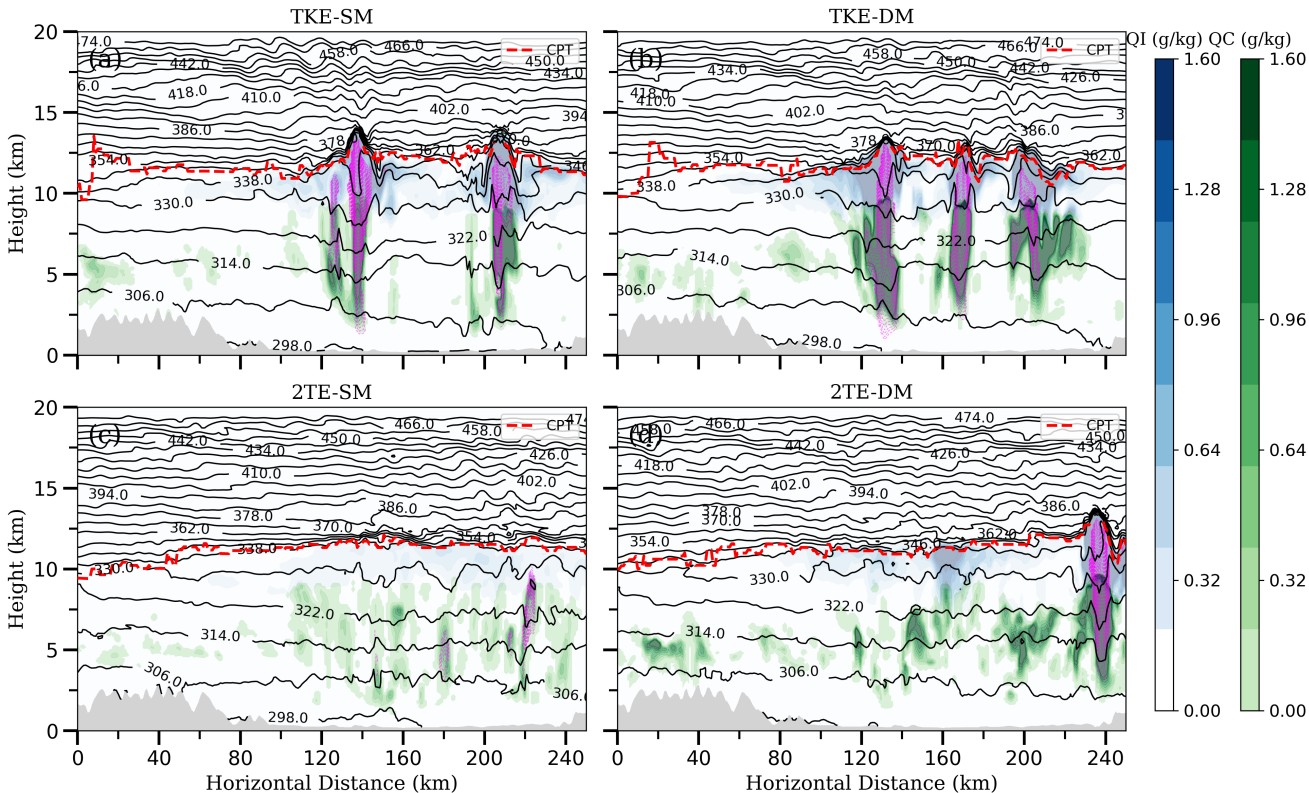

**Figure 12.** Same as Figure 11 but for QI (g/kg) and QC (g/kg). The magenta lines indicate regions where vertical wind exceeds $5\,\mathrm{m\,s^{-1}}$. The red dashed line in each panel show the altitude of the cold point tropopause (CPT), the altitude of minimum temperature in the troposphere

the hydrometeors and transport across the tropopause. As this analysis is based on a single case, further studies across diverse regions and conditions are needed to establish the broader applicability of these findings.

*Data availability.* The simulated ICON data presented in this study would be available on request. MSG/SEVIRI data are available from
the EUMETSAT (European Organisation for the Exploitation of Meteorological Satellites). The retrievals of CiPs, APICS are available on request from original data providers.



*Author contributions.* JS designed and conceptualized the research questions and goals. JQ and JS provided and helped with analysing ICON simulations. HKA performed the investigation, data analysis and wrote the original draft. LB and JM provided the observational data. JQ, JS, LB and JM reviewed and edited the draft. HKA edited the final manuscript

*Competing interests.* The authors declare no competing interests

*Acknowledgements.* This work was finanically supported by the Deutsche Forschungsgemeinschaft (DFG, German Research Foundation) – TRR 301 – Project-ID 428312742: "The tropopause region in a changing atmosphere, https://tpchange.de/" subproject B03 in collaboration with A02. The authors greatly acknowledge the computing time provided on the supercomputer levante-under project ID bb1096.



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
