# Peer review of "Evaluating Turbulent and Microphysical Schemes in ICON for Deep Convection over the Alps: A Case Study of Vertical Transport and Model–Observation Comparison"

_EGUsphere, 2025_

## Referee Comment (RC2)

**Review of *Evaluating Turbulent and Microphysical Schemes in ICON for Deep Convection over the Alps: A Case Study of Vertical Transport and Model-Observation Comparison* by Alladi et al.**

In this research article, a case study of deep convection over mountainous terrain is examined across 4 high resolution ICON model runs with variations in the utilized turbulence and microphysics schemes. Specifically, the authors utilize model configurations with single- and double-moment microphysics as well as with the default TKE turbulence scheme and the newer 2TE turbulence scheme. Thanks to the model's fine spatial resolution, deep convection is well-represented and does not require parameterization. The model runs are compared with satellite measurements, including retrievals of overshooting cloud tops, ice cloud cover, ice water path, and ice optical depth. The authors find that, in this case study, the choice of microphysics scheme and turbulence scheme have a significant impact on the intensity of convection and frequency of overshooting cloud tops.

The article's methods are sound and showcase interesting differences between the microphysics and turbulence schemes with respect to the case study at hand. I think it will be a valuable resource to the communities developing these parameterizations and exploring related processes. However, the paper requires some refinement before it is ready for publication. My comments on these issues below.

General comments:

1. Overall, the language and readability of this paper could use some improvement. I point out some specific examples below, but the paper could use a top-to-bottom re-evaluation in this area, ideally with a focus on sentence flow, grammar, and spelling. In some cases, like L41-47 and L341-343, relevant recent findings are described but not well-connected to the research at hand or the preceding sentence, leading to a stilted flow that doesn't lead the reader down a logical path. Along with the need for the described overall improvements, these specific passages should be revisited.

2. The impacts of this paper should be elaborated upon further in the Conclusions section. In its current state, the structure of the Conclusions section feels clipped, mostly summarizing the key results without too much discussion of the broader impacts. As you state, the analysis is based on a single case, and further studies across diverse regions and conditions are needed to contextualize the results. I'm sympathetic to that. But based on these results, do you have any preliminary suggestions as to what sort of experiments this research will motivate? Are there specific areas of weakness in the model that have been exposed that need to be

explored in future work? Were any aspects of your findings unexpected? How do these results contribute to the development and evaluation of the broader ICON framework?

3. You cite Gettelman et al., 2024, (hereafeter G24) a few times in the report, describing their result of increased IWP in double-moment (DM) schemes as a consequence of enhanced ice nucleation, reduced sedimentation of smaller cloud ice particles, and enhanced accretion of ice to snow (these effects are also discussed more explicitly in L271-277). While this is a useful citation and comparison, and I wouldn't necessarily expect the effects to be different, I think a little more work needs to be done to make adequate comparison between the single-column, Southern Ocean study of G24 to the authors' deep convective case in complex terrain over Western Europe. Are you able to examine some of the causal pathways towards higher IWP that are described in G24? I'm not trying to suggest that you perform a whole other set of simulations exploring the sensitivity of various parameterization setups as they do in G24- that seems like an unreasonable increase in scope. But additional evidence linking these conclusions would be useful. Are you able to compare upper-level sedimentation and aggregation between your different model outputs? What about graupel amount, which is relatively similar in the results from G24 but is expected to be more different in a deep convection case (such as the one presented here)? Some of these results can be seen by the reader, qualitatively, by looking at Figure 12, but it's not explicitly discussed in the text. Overall, closer examination of the process differences between the SM and DM runs would be helpful for understanding the causal root of those differences and contextualizing this work in the overall literature.

Specific comments:

Figures 4, 6, 7: Due to the formatting of the linestyles and the legend, it is challenging to distinguish between SM and DM model runs. For instance, here's the Figure 7 legend:

[Figure]

With close examination, you can see the slightly shorter line length that the -DM runs have compared to the -SM runs (marking them as dashed lines), but that detail is difficult to discern. I suggest you change the formatting of these legends to make the difference more obvious.

Figure 4: Extremely minor note, but your time axis for this diurnal cycle plot uses zero-padded single digits (e.g., "03" instead of "3") for the tick labels. You may want to consider matching the formatting of your other diurnal plots (Figs 6, 7, 8, and 10).

Figure 7: The high IWP from observations between 4 and 11 UTC is quite different from the model output here. Is this due to the advection of high clouds mentioned in L203 with respect to Fig. 4? If so, it may be useful to remind the reader of this detail. If not, it should be addressed.

Figure 9: For context, it would be useful to show the cross-section line (magenta line in Fig. 3) in these plots as well.

Figure 10: Are these time series for each day simulated (July 7, 8, and 9)? Because the plots are not labeled with the day they are representing and all analysis so far has been from July 8, the introduction of this new analysis is jarring. The temporal domain of each subplot needs to be specified (perhaps with labeling the columns of your subplots with the day in question?). Assuming this interpretation is correct, I am confused at the intent behind the introduction of these extra days of data. (b) and (e) are well motivated (assuming they are for July 8) because this is the case we have examined thus far. We have context for this data. But what additional information do we get from seeing the time series of CAPE and w for 7/7 and 7/9? Is the intent to show that the differences in model behavior are not limited to the circumstances present in the main case study day? Whatever the case, more rationale is needed to support the reason for the inclusion of the additional time series (or they should simply be cut). As-is, I find the 7/7 and 7/9 plots distracting and confusing more than helpful. For instance, in (a) there is a conspicuous difference in CAPE between 2TE-SM and 2TE-DM that is not seen as strongly in plots (b) and (c), but I have no context for what the causes for this discrepancy may be as this day is not examined or displayed anywhere else in the paper.

Figure 11 and 12: Units for potential temperature are not specified. Additionally, the subplot labels (a-d) are very difficult to see with the contour background. Consider moving these labels outside the plot. The same is true for the CPT legend in Figure 12.

L42: This sentence ("However, the deep convective…") is awkward and not grammatically correct.

L51: NWP is not defined.

L105: FAO is not defined.

L148-149: "CiPS" is not directly stated to be an acronym for the Cirrus Properties from SEVIRI algorithm.

L293: "strength" is misspelled.

L295-296: The line described here (and plotted in Figures 11 and 12) is the cross-section shown by the magenta line in Figures 3 and 5, and this is described as such in the Figure 11 caption. For the sake of readability, it would be useful to also state this explicitly in the text.

L335-337: Awkward sentence structure and incorrect capitalization of "Its", recommend reworking.

L336: ABL is not defined (although PBL is on L46)